# Anti-microRNA-21 Therapy on Top of ACE Inhibition Delays Renal Failure in Alport Syndrome Mouse Models

**DOI:** 10.3390/cells11040594

**Published:** 2022-02-09

**Authors:** Diana Rubel, Joseph Boulanger, Florin Craciun, Ethan Y. Xu, Yanqin Zhang, Lucy Phillips, Michelle Callahan, William Weber, Wenping Song, Nicholas Ngai, Nikolay O. Bukanov, Xingyi Shi, Ali Hariri, Hervé Husson, Oxana Ibraghimov-Beskrovnaya, Shiguang Liu, Oliver Gross

**Affiliations:** 1Clinic for Nephrology and Rheumatology, University Medical Center Goettingen, 37075 Goettingen, Germany; diana@rubel.de (D.R.); yiran6666@126.com (Y.Z.); 2Sanofi Research & Development, Cambridge, MA 02142, USA; joseph.boulanger@sanofi.com; 3Sanofi-Genzyme Research and Development, Framingham, MA 02118, USA; florin.craciun@sanofi.com (F.C.); ethan.xu@gmail.com (E.Y.X.); lucy.phillips@abbvie.com (L.P.); michelle.callahan@sanofi.com (M.C.); William.weber@takeda.com (W.W.); wenping.song@sanofi.com (W.S.); nicholas.ngai@sanofi.com (N.N.); nob158@gmail.com (N.O.B.); xingyishi001@gmail.com (X.S.); herve.husson@sanofi.com (H.H.); oxanab2020@gmail.com (O.I.-B.); 4Excision BioTherapeutics, San Francisco, CA 94111, USA; 5Department of Pediatrics, Peking University First Hospital, Beijing 100034, China; 6Abbvie Bioresearch Center, Worcester, MA 01605, USA; 7Takeda Pharmaceuticals, Cambridge, MA 02139, USA; 8Janssen Pharmaceuticals, Boston, MA 02115, USA; 9Novartis Institute for BioMedical Research, Boston, MA 02139, USA; 10Sanofi-Genzyme, Clinical Development, Cambridge, MA 02142, USA; Ali.Hariri@eloxxpharma.com (A.H.); shiguang.liu@sanofi.com (S.L.); 11Eloxx Pharmaceuticals, Watertown, MA 02140, USA; 12Dyne Therapeutics, Waltham, MA 02451, USA

**Keywords:** nephroprotection, renal fibrosis, type IV collagen, Alport syndrome, microRNA-21, podocytopathies, hereditary kidney diseases, kidney therapies

## Abstract

*Col4a3^−/−^* Alport mice serve as an animal model for renal fibrosis. MicroRNA-21 (miR-21) expression has been shown to be increased in the kidneys of Alport syndrome patients. Here, we investigated the nephroprotective effects of Lademirsen anti-miR-21 therapy. We used a fast-progressing *Col4a3^−/−^* mouse model with a 129/SvJ background and an intermediate-progressing F1 hybrid mouse model with a mixed genetic background, with angiotensin-converting enzyme inhibitor (ACEi) monotherapy in combination with anti-miR-21 therapy. In the fast-progressing model, the anti miR-21 and ACEi therapies showed an additive effect in the reduction in fibrosis, the decline of proteinuria, the preservation of kidney function and increased survival. In the intermediate-progressing F1 model, the anti-miR-21 and ACEi therapies individually improved kidney pathology. Both also improved kidney function and survival; however, the combination showed a significant additive effect, particularly for survival. RNA sequencing (RNA-seq) gene expression profiling revealed that the anti-miR-21 and ACEi therapies modulate several common pathways. However, anti-miR-21 was particularly effective at normalizing the expression profiles of the genes involved in renal tubulointerstitial injury pathways. In conclusion, significant additive effects were detected for the combination of anti-miR-21 and ACEi therapies on kidney function, pathology and survival in Alport mouse models, as well as a strong differential effect of anti-miR-21 on the renal expression of fibrotic factors. These results support the addition of anti-miR-21 to the current standard of care (ACEi) in ongoing clinical trials in patients with Alport syndrome.

## 1. Introduction

Alport syndrome (AS) is a hereditary disorder characterized by mutations in the genes encoding for the α3, α4 or α5 chain of type IV collagen, leading to defective glomerular basement membranes (GBMs). In the course of the disease, the thickening and splitting of the GBM are accompanied by hematuria, proteinuria and progressive renal fibrosis leading to end-stage renal failure [1,2].

Several animal models of AS have been established, one of which is the *Col4a3^−/−^* mouse model introduced by Cosgrove and colleagues [3]. Several treatments, including renin–angiotensin–aldosterone-system (RAAS) blockade, stem cells and anti-fibrotic therapy, have been evaluated in this model [4,5,6,7,8,9]. AS recently has become a treatable disease, as renal failure can be delayed by years and life expectancy improved by angiotensin-converting enzyme inhibitors (ACEi) [10,11,12]. However, new therapies beyond RAAS blockade are strongly needed to further delay renal fibrosis in AS [13].

MicroRNA-21 (miR-21) is upregulated in kidney diseases, functioning as a post-transcriptional regulator of gene expression modulating the tissue repair response after acute or chronic renal injury [14,15,16,17]. Different metabolic pathways, such as those regulated by the transcription factor PPARα, have been described as important targets for silencing by miR-21 activity in interstitial kidney fibrosis [15]. Gomez and colleagues described the effect of anti-miR-21 oligonucleotides on disease progression in *Col4a3^−/−^* Alport mice [18]. Anti-miR-21 therapy showed TGFβ-dependent antifibrotic and anti-inflammatory effects in the glomerulus as well as in the tubulointerstitium via enhanced PPARα/retinoid X receptor (PPARα/RXR) activity [18]. Further, reduced mitochondrial ROS production helped maintain the function of the tubular compartment. In conclusion, the previous data from *Col4a3^−/−^* Alport mice has raised hope that anti-miR-21 therapy might become a new therapeutic option in patients with Alport syndrome [15,18].

In the present study, we used Alport mice to inform the clinical setting for the ongoing HERA clinical trial (https://clinicaltrials.gov/ct2/show/NCT02855268, accessed on 4 December 2021) and future clinical trials involving the use of anti-miR-21 in humans with AS that already receive ACE inhibitor (ACEi) therapy [11,12,19]. To this end, two versions of the model and treatment regimens were employed. In a fast-progressing *Col4a3^−/−^* Alport mouse model with a 129/SvJ background, ACEi was started prophylactically at 4 weeks of age, while anti-miR-21 therapy was started at 6 weeks of age when kidney damage was already present. In a slower progressing (B6; 129SvJ-*Col4*α*3^tm1Dec^*) F1 model, which resulted from crossing *Col4a3^−/−^* Alport mice of 129/SvJ and C57BL/6J backgrounds, both therapies were started early at 5 weeks of age, prior to the manifestation of impaired kidney function. The effects on kidney function, pathology and survival of the individual and combined therapies were studied. RNA-seq analysis was used to investigate the pathways of gene expression that were similarly or differentially influenced by the two treatments.

## 2. Materials and Methods

### 2.1. Mouse Models and Treatments

*Col4a3^−/−^* mice (*Col4*α*3^tm1Dec^*) with a 129/SvJ background (Jackson Lab, Bar Harbor, ME, USA) were genotyped by PCR as described previously [4]. Animal study protocols were approved by the University Medical Center of Goettingen Institutional Animal Care and Use Committee (IACUC) and local German authorities. The animals were fed with R/MH diet ad libitum (V153x; Ssniff, Soest, Germany). ACEi-therapy was started pre-emptively in 4-week-old animals at 9 to 11 mg/L (to adjust for weight change) in drinking water. Ramipril (Delix^®^ Sanofi-Aventis, Berlin, Germany) was chosen as the ACEi because of previous studies in mice [4] and children [19] with AS. For this reason, the animal experiments simulated the most likely clinical scenario where AS patients would already be receiving ACEi medication as part of their standard of care before the addition of a therapy such as anti-miR-21. Anti-miR-21 Lademirsen—also known as RG-012, RG456070 or (SAR339375)—provided by Regulus Therapeutics Inc. (San Diego, CA, USA) was initiated in 6-week-old animals [18]. Anti-miR-21 (24.5 mg/kg) or PBS vehicle (Veh) were administered subcutaneously (SC) twice a week (BIW) using 5 ml/kg injection volume. Urine was collected upon spontaneous release while animal handling or by placing the mice in metabolic cages for 16 h at the indicated age.

Homozygous *Col4a3^−/−^* mice were divided into 4 groups:(1)Treated with Vehicle (Veh).(2)Treated with anti-miR-21 (Lademirsen).(3)Treated with Ramipril plus vehicle (ACEi).(4)Treated with Ramipril plus Lademirsen (ACEi-anti-miR-21).

A group of untreated wild-type (WT) *Col4a3^+/+^* control mice with a 129/SvJ background was also included in the studies. The duration of all therapies was not limited in time and lasted until the animals were found moribund or dead, and the date of the event was recorded. Four animals of each group were sacrificed at 7.5 and 9.5 weeks of age and their kidneys were further investigated using histological, immunohistological and Western blot techniques.

All animal experiments using the B6;129SvJ-*Col4*α*3^tm1Dec^* F1 model (F1-*Col4a3^−/−^*) and wild-type littermates B6;129SvJ- *Col4a3^+/+^* (WT) were conducted under protocols approved by Sanofi’s IACUC. Animals were generated as described previously [20]. The animals were fed PicoLab Rodent Diet 20 #5053 ad libitum (LabDiet, St Louis, MO, USA).

To measure the effect of Lademirsen, ACEi or the combination of both on lifespan in F1-*Col4a3^−/−^* mice, treatment was started at approximately 5 weeks of age and continued until the animals were found moribund or dead, and the date of the event was recorded. The groups were the same as for the studies performed with mice with the 129/SvJ background. Lademirsen (50 mg/kg) was administered by SC injection once weekly (QW) using 5 mL/kg injection volume to anti-miR-21 and ACEi + anti-miR-21 groups. Ramipril was provided ad libitum in drinking water at 11 mg/L to ACEi and ACEi + anti-miR-21 groups. Veh and ACEi groups received SC PBS injections of 5 mL/kg QW.

Blood was collected from WT, Veh, ACEi, anti-miR-21 and ACEi + anti-miR-21 groups at 5 (baseline), 7, 9, 11, 13 and 15 weeks of age, and from ACEi, anti-miR-21 and ACEi + anti-miR-21 groups at 17 and 19 weeks of age. For this, mice were transiently sedated under isoflurane anesthesia and approximately 200 µL blood was collected by retro-orbital eye bleed. The separated serum was assessed for renal function as described below.

To measure the effect of anti-miR-21 or ACEi on kidney pathology, determine the effect of anti-miR-21 on tissue miR-21 levels and perform kidney RNA-seq analysis, a satellite group of 8 males were treated as described above, sacrificed at 15 weeks of age and their tissues were either fixed or flash frozen.

### 2.2. Renal Function Evaluation

For the densitometric measurement of Coomassie blue-stained proteins in the urine of 129/SvJ mice, proteins were separated by 4–12% Novex Tris-Glycine PAGE (Life Technologies, Carlsbad, CA, USA), as described previously [4]. Serum samples, including blood urea nitrogen (BUN), were analyzed on a Cobas8000 Modular Analyzer Series (Roche Diagnostics, Mannheim, Germany). Renal function in the F1 model was assessed by measuring BUNas previously described [20].

### 2.3. Histology, Immunostaining and Quantitative Analysis

Kidneys from 129/SvJ mice were immersion-fixed as described previously [4]. Sections were cut to 3 µm thickness using a Reichert-Jung 2040 Autocut Microtome (Leica, Wetzlar, Germany). After fixation on glass slides, the sections were dewaxed in xylene and rehydrated in decreasing alcohol concentrations. Proteinase K was used to unmask laminin. Citrate buffer (pH 6.1) was used to unmask fibronectin. Anti-laminin 111 (ab11575, Abcam, Cambridge, UK) and anti-fibronectin (sc-6952, Santa Cruz Biotechnology, Dallas, TX, USA) antibodies were incubated overnight at 4 °C. Staining with secondary antibodies only served as negative controls. The sections were washed in TBST and incubated with a Cy3-conjugated secondary anti-rabbit antibody (611-104-122, Rockland Immunochemicals, Gilbertsville, PA, USA) or a TRITC-conjugated secondary anti-mouse antibody (T5393, Sigma-Aldrich, St. Louis, USA). After another step of washing in TBST and mounting, six different areas were documented for each kidney on a Zeiss Axiovert S100 TV microscope (Göttingen, Germany). Next, three independent, blinded volunteers scored the kidney sections as described previously [4], using a score from 0 (normal amount of laminin and fibronectin) to 3 (severely increased amount of deposition).

Kidneys from mice in the F1 model were fixed in 10% neutral-buffered formalin (Electron Microscopy Sciences, Harfield, PA, USA) for a minimum of 24 h before being embedded in paraffin. Sections were cut at a thickness of 5 µm and stained with hematoxylin and eosin (H&E), alpha smooth muscle actin (αSMA) or Masson’s trichrome. αSMA staining was performed after dewaxing and proteinase K treatment, as described for the kidney sections from 129/SvJ mice, using an αSMA antibody (M0851, Dako) followed by Rodent Block M and Mouse-on-Mouse HRP-Polymer (Biocare, MM510H) and DAB from the Bond Polymer Refine DAB Detection kit. All slides were scanned at 20X magnification using an Aperio ScanScope XT (Leica Biosystems, Wetzlar, Germany). All acquired images were evaluated for quality using the ImageScope viewer. Using the Aperio Image Analysis software (version 9.1), Genie, a pattern recognition algorithm, was applied to classify each scanned image into three categories: clear slide, tissue or other (unwanted artifacts, large vessels). A nuclear algorithm was applied to the areas previously defined as tissue by Genie. By defining the parameters that control cell segmentation and setting appropriate thresholds for positivity, the number of positive nuclei per area of analysis was calculated for each SMA; the percentages of SMA-positive nuclei were reported. For Masson’s trichrome staining, data were reported as percent blue area over total area. Histopathology scoring was performed by two independent nephropathologists who were blinded to the slide information. The pathology scoring criteria were modified from the criteria previously published by Remuzzi [21]. The scores were defined by the percentage of tissue exhibiting pathological changes out of the total reviewed area. Multiple fields were evaluated for each sample, and the mean value was used as the final score. The detailed pathological scoring criteria for glomerular and tubular pathology or interstitial inflammation and fibrosis were 0 (no pathological change was observed), 1 (lesions ≤ 25%), 2 (lesions 25–50%), 3 (lesions 50–75%) and 4 (lesions 75–100%) in the glomeruli, tubules or interstitium, respectively. F4/80 staining was performed as previously described [22].

### 2.4. Target Engagement and RNA-Seq Analyses

RNA was extracted from the kidneys of F1 model mice using a Qiagen miRNeasy kit for the target engagement studies or RNeasy for the RNA-seq analyses as described in the manufacturer’s manual. Samples were homogenized in lysis buffer using an Omni Bead Ruptor (Omni International, Kennesaw, GA, USA) set for 2 × 30 s beat cycles at 5.65 ms speed separated by a 30 s pause and then centrifuged at 15k× *g* at 4 °C.

Kidney miR-21 levels, normalized using miR-191, were measured using semi quantitative RT-PCR TaqMan assays (Assay ID 000397 and ID 000391, respectively, both from Life Technologies, Carlsbad, CA, USA). MiR-21 levels were calculated using the delta-delta CT method and expressed as a fold change relative to the WT control group ± SEM.

RNA-seq of total RNA samples was conducted using the Illumina HiSeq 2000 platform, producing 100 bp paired-end reads. All FASTQ files from the RNA-seq experiment were processed with QIAGEN OmicSoft Studio (version 9.0). Briefly, the sequence reads were mapped to the reference mouse genome (Genome Reference Consortium Build 38) by the OSA4 (OmicSoft Aligner, version 4) [23], with an average of 70 million paired-end reads aligned per library and >97% of the mapped reads in genic regions. Alignments were then quantified into FPKM (fragments per kilobase per million mapped reads) values at the gene level using the gene model from Ensembl R83 [24]. For outlier sample detection with principal component analysis (PCA) and hierarchical clustering analysis, the raw FPKM values were used directly. For differential expression analysis, Log2(FPKM + 1) values were used for one-way ANOVA on the effect of treatment with two contrasts: anti-miR-21 vs. vehicle and Ramipril vs. vehicle.

Prior to pathway enrichment analysis, the raw gene-level read counts were exported from OmicSoft Studio. The DESeq2 package (version 1.22.1) [25] within the R programming environment (version 3.5.1) was applied to generate the rankings of genes based on the differential expression patterns from three pairwise comparisons: (1) Veh vs. WT; (2) ACEi vs. Veh; and (3) anti-miR-21 vs. Veh.

Gene set enrichment analysis (GSEA) was used to identify differentially expressed pathways [26]. The GSEA analysis tool (version 3.0) was downloaded from the Broad Institute website (http://www.broadinstitute.org/gsea/index.jsp, accessed on 21 August 2018). The curated gene sets behind MetaCore pathway analysis are stored in the relational database, MetaBase (Clarivate Analytics, London, UK). MetaBase pathway gene sets (July 2018 version) were extracted with the metabaser R package (version 4.3.0), available through Sanofi-Genzyme’s subscription to the CBDD (Computational Biology Methods for Drug Discovery) program (Clarivate Analytics). More detailed examples of the naming of MetaBase pathways based on each pathway’s member genes were provided by Gomez et al. [18]. Pre-ranked GSEA was conducted with the MetaBase pathway definition file and the ranked gene lists from the DESeq2 analysis of the 3 pairwise comparisons [27].

To visualize the contributions of member genes to the most significantly regulated pathways, mean Log2(FPKM + 1) values were calculated for each gene within the four treatment groups: (1) WT, (2) Veh, (3) ACEi and (4) anti-miR-21. The expression profiles of all member genes were then plotted with the ggplot2 R package, where the genes were ordered by their mean Log2(FPKM + 1) values in the Veh group.

Data are available from the Gene Expression Omnibus with submission number GSE179938 (https://www.ncbi.nlm.nih.gov/geo/query/acc.cgi?acc=GSE179938, accessed on 4 December 2021).

### 2.5. Statistics

For studies in the 129/SvJ model, statistical analysis was performed using a one-tailed Student’s T test for the comparison of proteinuria and BUN. The scoring of the histological sections was analyzed by two-way tests of variance (ANOVA). A log-rank test was used for survival analysis.

For studies in the F1 model, a log-Rank test with multiple comparisons was performed to analyze survival. Each KO group was compared to one another. A two-way ANOVA-type with the factors treatment and week (repeated) was performed on BUN. This was followed by a contrast analysis for pairwise comparison between different groups at each week. In order to confirm disease progression, a two-way ANOVA-type with the factors genotype (wildtype vs. control) and week (repeated) was also performed on BUN. This was followed by a contrast analysis between wildtype and control at each week. All analyses were performed using Everst@t version 6.1 (Sanofi’s in-house general statistical package based on SAS) and a statistical analysis system; *p*-values less than 0.05 were considered significant. For target engagement, to determine if the groups contained outliers, Dixon’s test was performed. No outliers were determined at a 1% inference level. A normality test followed by a Levene test for equal variances was further performed. Both the normality and equal variance assumptions were not violated. Therefore, the groups were compared using an unpaired Student’s t-test.

All studies reported herein adhered to good laboratory practices established by the University Medical Center Goettingen, in Goettingen, Germany and Sanofi-Genzyme, in Framingham, USA.

## 3. Results

### 3.1. Effect of ACEi, Anti-miR-21 or the Combination of Early ACEi and Anti-miR-21 on Kidney Failure and Survival of AS Mice in 129/SvJ Background

Life expectancy until renal failure is the most important clinical endpoint for humans with chronic kidney disease. The median survival time of vehicle-treated *Col4a3^−/−^* mice with the 129/SvJ background was 10 weeks (Figure 1a,b and Appendix A). ACEi monotherapy with Ramipril significantly improved survival by 53% to 15.3 weeks compared to Veh (*p* < 0.05), while anti-miR-21 monotherapy improved survival by 14% to 11.4 weeks compared to Veh (*p* < 0.05). The combination therapy further improved the median survival time by 116% compared to the Veh group to 21.6 weeks, showing superiority over Ramipril and anti-miR-21 monotherapies.

Glomerular and tubular function was further evaluated by albumin and high molecular weight protein excretion quantified by densitometry analysis in 4.5-, 6.0-, 7.5- and 9.5-week-old mice. In week 7.5, albuminuria and high molecular weight proteinuria (mainly immunoglobulins) were more than 25-fold higher in Veh Alport mice compared to the wildtype mice. Compared to Veh Alport mice, anti-miR-21 monotherapy lowered high molecular weight proteinuria by more than 50%. ACEi and ACEi + anti-miR-21 lowered high molecular weight proteinuria by approximately 90% (all data compared to Veh Alport mice at week 7.5). Due to advanced kidney fibrosis and ongoing oliguric renal failure, the comparison of proteinuria in 9.5-week-old Alport mice was less meaningful. However, all treatment combinations, including anti-miR-21 monotherapy, had a substantial effect on high molecular weight proteinuria as a sign of glomerular damage. All treatment combinations with Ramipril (ACEi and ACEi + anti-miR-21) also showed a considerable effect on albuminuria (Figure 1c, Appendix A).

The loss of renal function was determined by the values of blood urea nitrogen (BUN) in 7.5- and 9.5-week-old mice with different treatment modalities (Figure 1d, Appendix A). The BUN in 7.5-week-old Veh Alport mice did not differ significantly from anti-miR-21-treated mice; however, anti-miR-21 therapy significantly improved BUN levels in 9.5-week-old mice. The BUN in all ACEi-treated mice (ACEi and ACEi + anti-miR-21) significantly improved compared to the Veh Alport mice. While ACEi preserved kidney function, as measured by BUN, anti-miR-21 did not provide the same protection in animals with already established disease. However, the combination of ACEi and anti-miR-21 lowered BUN values to 42.0 ± 5.4 mg/dl (7.5 weeks) and to 60.6 ± 3.6 mg/dl (9.5 weeks).

### 3.2. Effect of ACEi, Anti-miR-21 or the Combination of Early ACEi and Anti-miR-21 on Kidney Pathology in AS Mice with a 129/SvJ Background

The extracellular matrix accumulation in kidney sections from 7.5- and 9.5-week-old mice was scored in an observer-blinded manner (Figure 2a–f, Appendix A). In WT mice, extracellular laminin deposition was very low, with a glomerular score of 0.5 ± 0.1 (Figure 2c) and a tubulo-interstitial score of 0.6 ± 0.1 (Figure 2f). In contrast, Veh Alport mice (Figure 2a) showed a very high deposition of laminin (glomerular and tubulo-interstitial score at week 7.5 and 9.5: *p* < 0.001 vs. WT) as did anti-miR-21-treated mice (Figure 2b). In contrast, ACEi + anti-miR-21 (Figure 2e) treatment markedly reduced the glomerular matrix accumulation by approximately 43% (*p* < 0.001 vs. Veh), as did ACEi alone (Figure 2d) (*p* < 0.001 vs. Veh). Combination therapy reduced tubulo-interstitial matrix deposition by about 40% in week 9.5 (ACEi + anti-miR-21; *p* < 0.001 vs. Veh).

Renal scar tissue formation (Figure 2g–l, Appendix A) was evaluated in WT mice, n showed low glomerulosclerosis (Figure 2i) and tubulo-interstitial fibrosis scores (Figure 2l). In contrast, Veh Alport mice (Figure 2g) showed major glomerular and tubulo-interstitial scarring (*p* < 0.001 vs. WT). Anti-miR-21 (Figure 2h) significantly reduced the glomerular scar tissue formation and reduced tubulo-interstitial fibrosis (*p* < 0.001 vs. Veh). ACEi + anti-miR-21 (Figure 2k) even further reduced glomerular scarring by more than 70%. ACEi + anti-miR-21 therapy also reduced tubulo-interstitial fibrosis by more than 60%. Similarly, ACEi monotherapy reduced tubulointerstitial fibrosis (Figure 2j).

### 3.3. Effect of ACEi, Anti-miR-21 or the Combination of ACEi and Anti-miR-21 on Survival and Renal Function in Moderately Slow-Progressing Alport Mice

*Col4a3^−/−^* mice were generated as an F1 hybrid cross between 129-*Col4*α*3^tm1Dec^*/J heterozygous males and C57BL/6J *Col4a3* heterozygous females (F1-*Col4a3^+/-^*), as previously described [20]. This F1 model with intermediate severity was chosen to allow longer windows to test therapeutic interventions for AS. Anti-miR-21, ACEi or the combination of anti-miR-21 and ACEi treatments were introduced early in disease progression at 5 weeks of age, prior to manifestation of impaired kidney function with increased BUN.

The median survival time of vehicle-treated F1-*Col4a3^−/−^* male mice was 16.5 weeks. ACEi monotherapy with Ramipril significantly improved survival to 18.8 weeks (*p* = 0.0003), while anti-miR-21 monotherapy resulted in even longer survival to 21.9 weeks (*p* < 0.0001). The combination therapy further improved the median survival time to 23.6 weeks, showing superiority over both Ramipril monotherapy (*p* = 0.0002) and anti-miR-21 monotherapy (*p* = 0.0034) alone (Figure 3a,b, Appendix A). To assess renal function, serum samples were collected biweekly from 5 weeks of age until the end of the study and measurements of BUN were performed. Compared with vehicle-treated F1-*Col4a3^−/−^* mice, both ACEi and anti-miR-21 therapies were effective in delaying the progression of kidney disease, which manifested as slower increases in BUN (Figure 3c, Appendix A). At the doses used, anti-miR-21 was more potent than ACEi monotherapy. The combination therapy showed an additive effect on BUN levels over the anti-miR-21 monotherapy until up to approximately 19 weeks of age.

### 3.4. Effect of Anti-miR-21 or ACEi on Kidney Pathology in F1-Col4a3^−/−^ Mice

The F1 hybrid model of AS with the *Col4a3* homozygous mutation develops kidney disease with the first significant increase in renal miR-21 starting at approximately 9 weeks of age, correlated with increased BUN. By 15 weeks of age, the kidneys show increased miR-21 in cortical renal tubules [20]. A separate cohort of F1- *Col4a3^−/−^* male mice were treated with anti-miR-21 or ACEi and sacrificed at 15 weeks of age with the purpose of assessing kidney pathology. A histopathological analysis of the kidneys, including qualitative and semi-quantitative assessment of the presence of glomerular pathology, tubular pathology and interstitial inflammation and fibrosis, were conducted in an observer-blinded manner. Both treatments slowed kidney function decline, as measured by BUN, with anti-miR-21 showing better protection compared to ACEi administration (Figure 4a, Appendix A).

To assess the involvement of miR-21 in the treatment effect of anti-miR-21, microRNAs were isolated from the kidney tissue and miR-21 expression was measured by qRT-PCR. MiR-21 expression was significantly increased (~7.7-fold) in vehicle F1-*Col4a3^−/−^* mice compared to WT mice (Figure 4b, Appendix A). The use of ACEi monotherapy in F1-*Col4a3^−/−^* mice lead to a significant decrease in kidney miR-21 expression (~3.4-fold over WT), however, not to the extent observed with anti-miR-21 therapy, which resulted in a reduction in kidney miR-21 expression to levels that were close to those in the WT group.

The analysis of H&E-stained kidney sections from Veh F1-*Col4a3^−/−^* mice showed severe glomerular and tubular pathology, accompanied by interstitial inflammation in the renal cortex compared to WT mice (Figure 4c top panels). Semi-quantitative analyses confirmed that Veh F1-*Col4a3^−/−^* mice displayed significantly increased glomerular pathology, tubular pathology and interstitial inflammation relative to the WT group, all reaching statistical significance (*p* < 0.05) (Figure 4d, Appendix A). Anti-miR-21 or ACEi therapies ameliorated renal pathology relative to Veh controls, with anti-miR-21 showing significantly improved glomerular and tubular pathology compared to the ACEi group (Figure 4c top panels, Figure 4d). Semi-quantitative analyses confirmed that anti-miR-21-treated F1-*Col4a3^−/−^* mice displayed significantly reduced glomerular pathology, tubular pathology and interstitial inflammation relative to the Veh group with *p* < 0.05 (Figure 4d, Appendix A).

In comparison, ACEi-treated F1-*Col4a3^−/−^* mice only displayed a significant reduction in tubular pathology and interstitial inflammation (*p* < 0.05) compared to the Veh group, with no effect on glomerular pathology. The quantification of the number of macrophages present in kidney sections (F4/80 immunohistochemistry staining) was used to further investigate the inflammatory response. The results confirmed increased inflammation in the Veh group compared to the WT group, as well as a reduction in inflammation in both treated groups compared to Veh group (Appendix A). Masson’s trichrome-stained kidney sections showed that Veh F1-*Col4a3^−/−^* mice had moderately severe interstitial fibrosis (black arrow heads) in the renal cortex (Figure 4c, middle panels), which was significantly increased compared to the WT group with *p* < 0.05 (Figure 4d, Appendix A). Anti-miR-21 or ACEi therapies showed the significant amelioration of interstitial fibrosis relative to the Veh group (Figure 4c), which was confirmed upon semi-quantitative analyses with *p* < 0.05 for both treated groups. Furthermore, anti-miR-21 treatment showed the significant amelioration of interstitial fibrosis compared to ACEi (Figure 4d, Appendix A). Kidney sections from the Veh group showed an increase in αSMA labeling in the tubular and glomerular interstitium compared to the WT group. The staining of kidney sections from the anti-miR-21 or ACEi groups showed a reduction in αSMA labeling in the interstitium (Figure 4c, bottom panels). Semi-quantitative analyses demonstrated a significant increase in αSMA in Veh vs. WT, ACEi and anti-miR-21 groups (*p* < 0.05). These improvements in the microscopic renal pathology upon anti-miR-21 or ACEi therapy were correlated with an improvement in BUN data.

### 3.5. Anti-miR21 Treatment Led to Larger Magnitudes of Expression Normalization of Differentially Expressed Genes than Ramipril Treatment

To better understand the mechanisms by which ACEi and anti-miR-21 correct the kidney defects of F1-*Col4a3^−/−^* mice, we performed next-generation RNA sequencing (RNA-seq) in kidney samples from WT, Veh, ACEi and anti-miR-21 groups at 15 weeks of age.

The first three principal components derived from the full RNA-seq data captured 53.5% of the total variance and showed good between-group separations (Figure 5a). With a fold-change threshold of ±1.2 and a *p*-value cutoff of 0.01, one-way ANOVA revealed 11,258 differentially expressed genes when comparing the Veh and WT groups (not shown). When comparing anti-miR-21-treated and Veh mice, 9,404 genes showed differential expression (Figure 5b). ACEi treatment resulted in 6869 differential genes, 6416 of which also appeared in the list from the anti-miR-21 treatment group (Figure 5b), indicating that the two treatments modulate many common pathways. The hierarchical clustering of these 6416 treatment-related differential genes highlighted the transcriptional responses to the drug treatments. Three of the six anti-miR-21-treated and one of the three ACEi-treated F1-*Col4a3^−/−^* mice responded to the treatment in a manner that brought the overall expression profiles close to that of the WT animals. For most of the differentially expressed genes, anti-miR-21 treatment led to larger magnitudes of normalization toward WT levels than did ACEi treatment (Figure 5c).

### 3.6. Anti-miR-21, but Not ACEi, Significantly Normalized Gene Expression Patterns in Renal Tubulointerstitial Fibrosis

A major limitation of many pathway overrepresentation analysis (ORA) methods (e.g., Ingenuity and MetaCore pathway analyses) is that they do not indicate which phenotype (condition) is associated with each overrepresented pathway [28]. The gene set enrichment analysis (GSEA) algorithm is a representative functional class scoring (FCS) method that overcomes this limitation [29].

All genes in a transcriptome are ranked by their correlation with one of the two phenotypes in a given pairwise comparison; then, the list is compared to sets of genes in a pathway database, linking pathway enrichment to a phenotype. More highly correlated genes in a given pathway would lead to a larger absolute value of the normalized enrichment score (NES) and a smaller significance *p*-value based on the null distribution of the Kolmogorov–Smirnov statistic [26]. GSEA of F1-*Col4a3^−/−^* renal transcriptomes compared with WT using the MetaBase pathway database identified “renal tubulointerstitial injury” (defined through studies of lupus nephritis) as one of the most significantly upregulated pathways. Pathways associated with immune responses and cytokine-induced fibroblast/myofibroblast migration were also upregulated in Alport mouse kidneys compared to WT (Table 1). Among the pathways most significantly downregulated in F1-*Col4a3^−/−^* kidneys, the suppression of oxidative phosphorylation and mitochondrial function were the most prominent (defined through studies of neurodegenerative diseases).

To better understand the mechanisms of action of anti-miR21 and ACEi in the Alport mouse kidneys, we also conducted GSEA pathway analysis on the renal transcriptomes from the Veh, anti-miR-21 or ACEi groups (Table 2 and Table 3). “Renal tubulointerstitial injury” was the MetaBase pathway most downregulated by anti-miR-21 treatment (Table 3), while its downregulation by ACEi was only ranked at No. 24 among the MetaBase pathways (data not shown). Both ACEi and anti-miR-21 were able to reverse-modulate other top-ranking pathways regulated by Alport pathogenesis, including “oxidative phosphorylation” and “mitochondrial dysfunction” (Table 2 and Table 3).

Given the functional relevance of the “renal tubulointerstitial injury” pathway, we further examined its member gene expression profiles in all four groups of mouse kidneys. The upregulation of this pathway in Alport mouse kidneys appeared to be driven by TGFβ1 (Tgfb1), fibronectin (Fn1), vimentin (Vim), multiple isoforms of α and β integrins (Itga1, Itgb1, and Itgb2) and fibrogenic collagens (Col1a1, Col1a2, Col3a1). Anti-miR-21 consistently showed larger magnitudes of expression reversal towards WT mice than ACEi did (Figure 6). In addition, anti-miR-21 also attenuated the downregulation of genes in the mitochondrial function pathway to a greater extent than ACEi did (Figure 7).

## 4. Discussion

Our experiments demonstrated that human patients with AS and progressive disease despite RAAS blockade may benefit from anti-miR-21 therapy in addition to high-dose ACE inhibition. Two different targets of add-on therapy with anti-miR-21 were identified: (1) the glomerular and (2) the tubulointerstitial compartment of the kidney. We postulated that in Alport pathogenesis, the inability of the podocyte to produce α3/4/5 (IV) collagen results in an overreaction of the podocyte, causing further glomerular damage. Calming of the podocyte via its local RAAS system by ACE inhibitors [4] or the loss of type IV collagen receptors [20] delays damage in AS. Similarly, anti-miR-21 may reduce the oxidative stress of the podocyte, as miR-21 targets the redox metabolic pathway, which correlates with oxidative kidney damage [15,18]. The preservation of glomerular function by anti-miR-21 and ACEi + anti-miR-21 is in line with the reduction in high molecular weight proteinuria.

Anti-miR-21 therapy further protected the tubulointerstitial compartment from scar-tissue formation. The anti-miR-21 tested here is a 19-base oligonucleotide with phosphorothioate (PS) inter-nucleotide links and 2-methoxyethyl or constrained ethyl base modifications. The addition of these PS modifications increases protein binding, which reduces renal clearance. The kidneys and liver are the major sites of accumulation of the compound; in the kidney, the compound is mostly accumulated in the renal tubule epithelium [18]. The majority of renal accumulation occurs through uptake in the capillary network, as shown for phosphorothioated-derived oligos [30]. In Alport pathogenesis, the weakened GBM causes progressive albuminuria and may enable anti-miR-21 compounds to pass the GBM barrier and enhance uptake in tubular cells. Therefore, the albumin-binding of anti-miR-21 compounds might be an advantage in chronic kidney diseases. Regardless of the ultimate mechanism of uptake, the tubule epithelia have the highest levels of uptake and are in the greatest need of protection from damage in AS [2,13]. Interestingly, our RNA-seq data showed a superior capability of anti-miR-21 at the doses tested during the course of these studies (when compared to ACEi) to normalize profibrotic and inflammatory pathways towards the wildtype level. We performed GSEA pathway analysis to better understand the effects of ACEi and anti-miR-21 on the renal transcriptome in the context of our F1 Alport mouse model. The comparison of F1-*Col4a3^−/−^* kidneys with WT kidneys revealed many similar top-ranking pathways to an earlier study of 129/SvJ *Col4a3^−/−^* mice using MetaCore pathway analysis [18]. A key finding of the current analysis was that the “renal tubulointerstitial injury” pathway, which is upregulated in F1-*Col4a3^−/−^* mice and includes several genes involved in kidney fibrosis, such as Tgfb1, fibronectin, vimentin, integrins and collagens, was differentially affected by anti-miR-21 and ACEi treatments. There was a higher level of expression normalization seen with anti-miR-21 compared with ACEi, which suggests a mechanistic difference between the two treatments. There were many pathways that were normalized towards WT levels by both anti-miR-21 and ACEi treatments, suggesting that perhaps they influence some common key regulators. Such was the case with the “mitochondrial dysfunction” pathway; however, closer analysis revealed that even in this case, at the doses tested here, anti-miR-21 had a stronger effect than ACEi on the expression levels of the genes involved in this pathway. While RNA-seq and GSEA were not performed for the combination treatment of ACEi and anti-miR-21, since many of the pathways identified for the individual treatments were common and considering the additive effects of the treatments on clinically relevant markers in two models of different severity and at different dosages, the combination treatment likely allows further expression normalization of the genes involved in the individual pathways to levels that could not be achieved with the maximum dose of an individual treatment. It is important to acknowledge that both preclinical animal models, the fast-progressing and the intermediate-progressing, were used independently in order to inform the clinical setting for the ongoing HERA clinical trial (https://clinicaltrials.gov/ct2/show/NCT02855268, accessed on 4 December 2021). In the context of the Alport patient group (Alport Syndrome Foundation) serving as sponsor of this preclinical study, we intended to enrich the planning of the clinical trial in humans. This also explains the different experimental settings used by both groups, including the lower dose of Ramipril used in the intermediate model.

Our study was able to demonstrate that anti-miR-21 can act as an additive to RAAS blockade to improve outcomes in Alport syndrome. Ramipril is known to delay glomerular and tubulointerstitial fibrosis and to prolong lifespan in Alport mice [4] as well as in humans [10]. For this reason, in the rapid progression model, we used early-on therapy with the maximum tolerated dose of ACEi in 129/SvJ background mice before initiating the anti-miR-21 therapy. Compared to previous reports, the anti-miR-21 monotherapy was applied quite late in the course of Alport disease in the rapid animal model [18], with 6-week-old mice corresponding to chronic kidney disease (CKD) stage CKD-2 or -3 in human Alport patients [10]. This point in time of overt proteinuria is the most relevant for the initiation of nephroprotective therapy in all other chronic kidney diseases leading to progressive fibrosis. While the effect of anti-miR-21 monotherapy on the lifespan of Alport mice until death from end stage renal failure was limited, anti-miR-21 monotherapy still showed a profound effect on renal function (BUN). Due to the rapid rate of progression to end stage renal disease in this model, it is possible that it could not significantly improve survival when administered alone, while still working synergistically to improve the effects of early-start ACEi therapy. In the slower progressing F1 mouse model, anti-miR-21 was applied early in the course of the disease, and in this case kidney pathology, function and survival were all significantly improved. Anti-miR-21 therapy initiated even later in the disease progression in this slower model was shown to be protective, although to a lesser extent [20]. This supports the further investigation of the administration of anti-miR-21 therapy early in AS disease progression.

## 5. Conclusions

In conclusion, adding anti-miR-21 to early ACEi therapy (anti-miR-21 + ACEi) delayed the progression of renal fibrosis and prolonged lifespan. Our study gives evidence of the nephroprotective effect of anti-miR-21 added to early-on ACE inhibition. Taken together with our previous report that miR-21 expression was found to be significantly elevated in kidney specimens from patients with AS and correlated with increased proteinuria, decreased renal function and the increased severity of kidney pathology [20], anti-miR-21 presents a promising new therapy for patients with AS, and is currently being investigated in the HERA trial (https://clinicaltrials.gov/ct2/show/NCT02855268, accessed on 4 December 2021). The mouse data presented in this manuscript give reason to hope that the combination of ACE inhibition and the inhibition of miR-21 will be synergistic. Therefore, this effect may further delay renal failure in humans over that previously observed with ACE inhibition alone, and could become an important piece of long-term therapy for AS [31].

## Figures and Tables

**Figure 1 cells-11-00594-f001:**
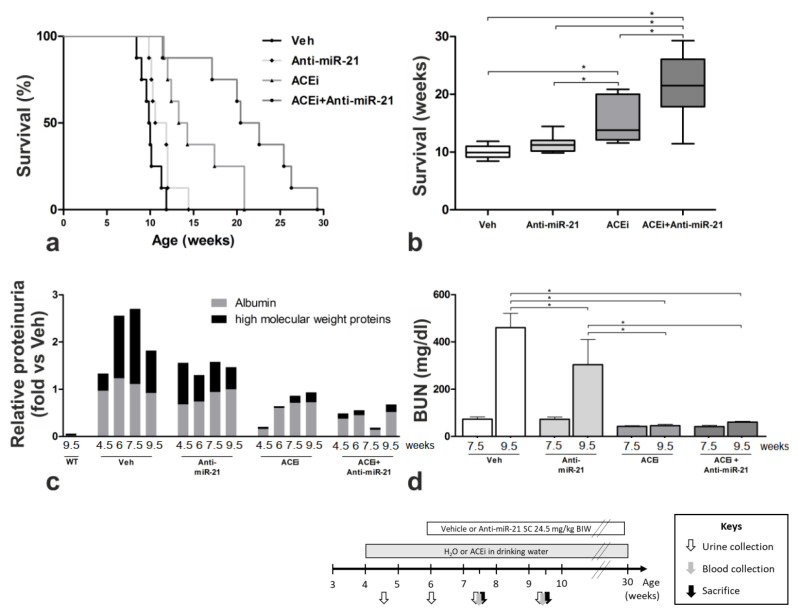
Effect of anti-miR21 on top of pre-emptive ACEi therapy in Alport mice of 129/SvJ genetic background (see schematic illustration of animal study design). (**a**) Lifespan of Alport mice displayed as a Kaplan–Meier-survival curve (*n*= 8) and (**b**) as a box plot (*n*= 8). (**c**) Proteinuria (*n* = 3) and (**d**) blood urea nitrogen (BUN) in different stages of progressive renal disease (anti-miR-21 *n* = 2; all other groups *n* = 3; data displayed as mean ± SEM). The excretion of albumin and high molecular weight proteins represents the glomerular damage due to the GBM-defect in Alport syndrome. BUN correlates inversely to renal function and was close to normal values in ACEi and ACEi + anti-miR-21-treated Alport mice. * *p* < 0.05.

**Figure 2 cells-11-00594-f002:**
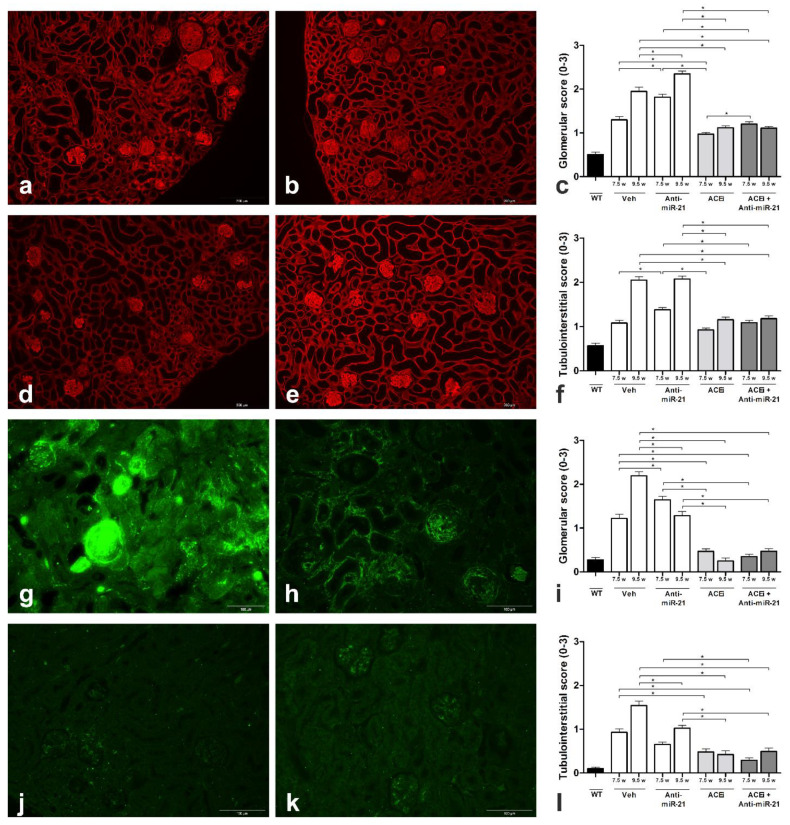
(**a**–**f**) Glomerular and tubulo-interstitial accumulation of extracellular matrix shown by immunofluorescence staining of laminin 111. At 9.5 weeks of age, Veh (**a**) and anti-miR-21-treated Alport mice (**b**) displayed severe glomerular and tubular matrix deposition. ACEi (**d**) and ACEi + anti-miR-21-treated Alport mice (**e**) had fairly preserved glomerular and tubulointerstitial architecture. Score of glomerular (**c**) and tubulo-interstitial (**f**) extracellular matrix accumulation. (**g**–**l**) Glomerular and tubulo-interstitial scar tissue formation represented by immunofluorescence staining of fibronectin. At 9.5 weeks of age, Veh Alport mice (**g**) had severe glomerular and tubular scarring. Anti-miR-21-treated Alport mice (**h**) showed significantly reduced scar tissue formation at the latest stage (9.5 w). ACEi-treated Alport mice (**j**) and ACEi+ anti-miR-21-treated Alport mice (**k**) had a glomerular fibrosis score close to the wildtype controls and an almost preserved tubulointerstitial architecture. Score of glomerular scar tissue formation (**i**) and tubulo-interstitial fibrosis (**l**). The magnification was 200X. WT—wildtype; *n* = 3; data displayed as mean ± SEM; * *p* < 0.05.

**Figure 3 cells-11-00594-f003:**
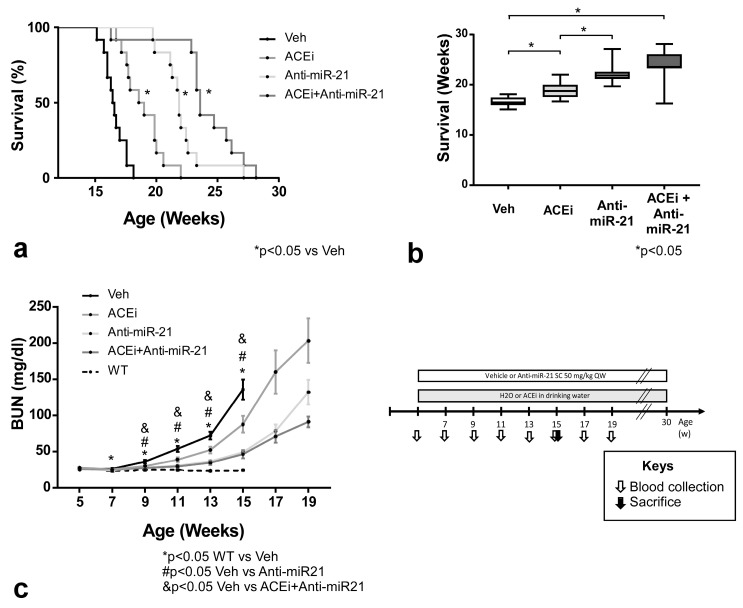
Effect of ACEi, anti-miR-21 or the combination of ACEi and Anti-miR-21 on survival, renal function and kidney miR-21 levels in F1-Col4a3^−/−^ male mice (see schematic illustration of study design). (**a**) Kaplan–Meier survival curve of Veh, ACEi, anti-miR-21 or ACEi + anti-miR-21-treated F1-Col4a3^−/−^ mice. Treatment was initiated at 5 weeks of age, *n* = 12. * *p* < 0.05 by log-rank test vs. Veh group. (**b**) Median survival shown as a box-plot graph. * *p* < 0.05. (**c**) Mean serum BUN ± SEM over time in WT, Veh, ACEi, anti-miR-21 and ACEi + anti-miR-21 groups. *p* values as shown in figure legend.

**Figure 4 cells-11-00594-f004:**
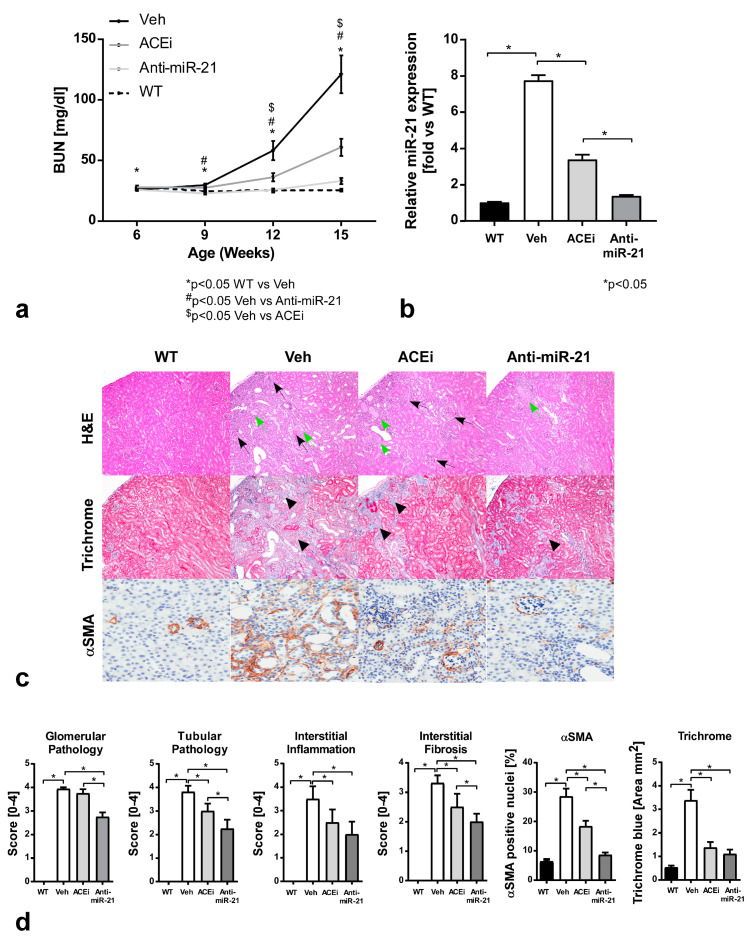
Effect of ACEi or anti-miR-21 on renal pathology and miR-21 expression in F1-*Col4a3^−/−^* male mice. (**a**) Mean serum BUN ± SEM over time in WT, Veh, ACEi and anti-miR-21 groups. *p* values as shown in figure legend. (**b**) Relative miR-21 expression measured by quantitative RT PCR in whole kidney of WT controls and F1-*Col4a3^−/−^* mice Veh, ACEi and anti-miR-21 groups. *n* = 6–7 * *p* < 0.05. (**c**) Representative kidney histopathology sections from WT and F1-*Col4a3^−/−^* mice treated with Veh, ACEi and anti-miR-21. Top—hematoxylin and eosin staining (H&E) showing tubular pathology (atrophy degeneration/regeneration represented by black arrows and intratubular protein casts by green arrows); middle—Masson’s trichrome staining (trichrome), blue Masson’s staining indicates interstitial fibrosis, marked by black arrowheads; bottom—αSMA staining showing SMA (brown) deposition in the interstitium. (**d**) Scores for glomerular pathology, tubular pathology, interstitial inflammation and interstitial fibrosis were determined as described in the Materials and Methods section. αSMA and trichrome quantitation were expressed as the percentage of positive cells relative to the nuclei and total surface area in mm^2^, respectively. * *p* < 0.05, as noted in figure.

**Figure 5 cells-11-00594-f005:**
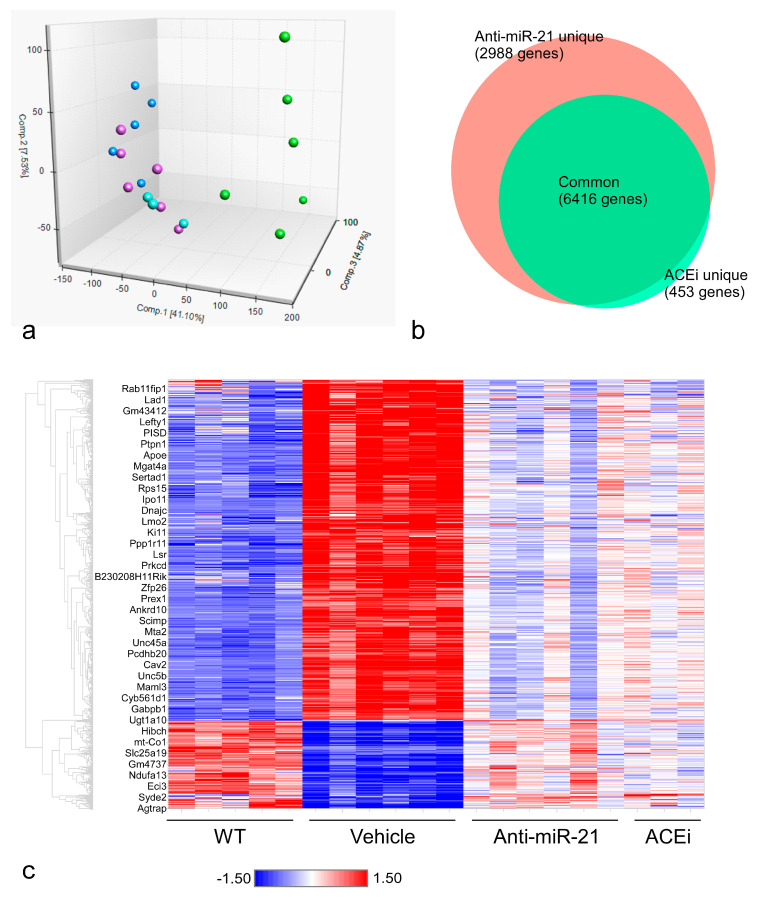
Renal transcriptomic analysis of wild-type and F1-*Col4a3^−/−^* mice. (**a**) Distribution of mouse kidney samples using the first three principal components computed from the RNA-seq FPKM data. Each dot represents one mouse sample, color-coded according to the genotype and treatment group (blue: WT controls; green: vehicle-treated F1-*Col4a3^−/−^*; purple: anti-miR-21-treated F1-*Col4a3^−/−^*; cyan: ACEi-treated F1-*Col4a3^−/−^*). (**b**) Venn diagram of the number of differentially expressed genes in F1-*Col4a3^−/−^* mice from the pairwise comparisons of “anti-miR-21 vs. Veh” and “ACEi vs. Veh”. (**c**) Hierarchical clustering of 6416 common treatment-related differential genes across all mouse kidney samples in the study. Kidney samples represented by columns of the clustering heatmap are sorted by the grouping of corresponding genotype and treatment (same color code as in panel (**a**)).

**Figure 6 cells-11-00594-f006:**
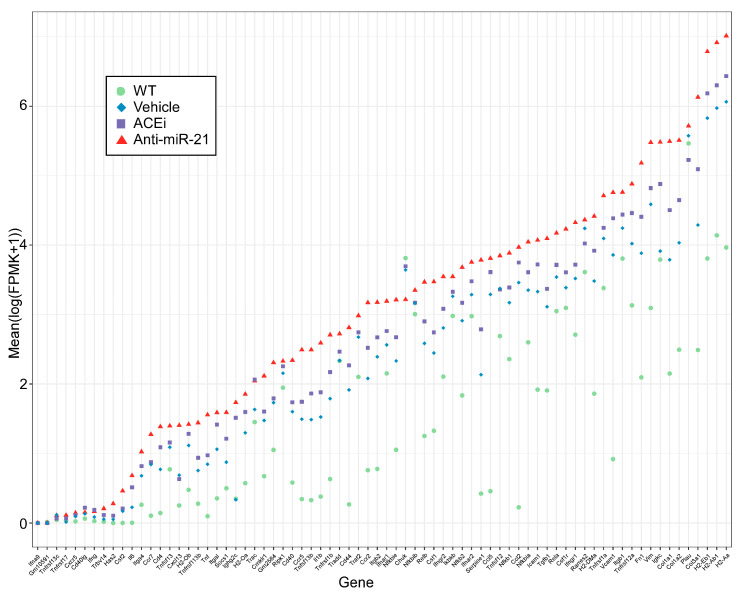
Member gene expression profiles from the MetaBase pathway “Renal tubulointerstitial injury in lupus nephritis” (curated by Clarivate Analytics). The Y axis represents Log2(FPKM + 1) values in each of the four animal groups and the genes on the X axis are ordered by their mean expression values in the vehicle-treated F1-*Col4a3^−/−^* group.

**Figure 7 cells-11-00594-f007:**
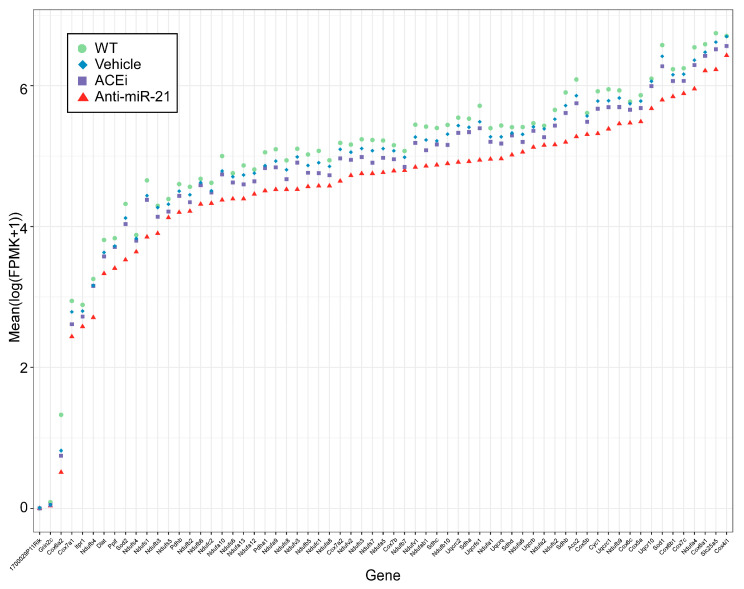
Member gene expression profiles from the MetaBase pathway “Mitochondrial dysfunction in neurodegenerative diseases” (curated by Clarivate Analytics), with the same plotting parameters as in Figure 6.

**Table 1 cells-11-00594-t001:** The most significantly regulated pathways from GSEA of Veh compared with WT mouse kidneys.

Top 5 Upregulated Pathways in Alport Mice	Size	ES	NES
Immune response: T-cell co-signaling receptors	62	0.81	2.28
Chemokines in inflammation in adipose tissue and liver	50	0.82	2.28
Renal tubulointertitial injury in lupus nephritis	74	0.78	2.24
Breakdown of CD4 and T cell peripheral tolerance in type 1 diabetes mellitus	70	0.77	2.22
Cytokine-induced fibroblast/myofibroblast migration and extracellular matrix production	42	0.82	2.21
**Top 5 Downregulated Pathways in Alport Mice**	**Size**	**ES**	**NES**
Oxidative phosphorylation	77	−0.84	−3.35
Ubiquinone metabolism	38	−0.84	−2.87
Mitochondrial dysfunction in neurodegenerative diseases	91	−0.67	−2.74
Mechanism of Pioglitazone/Rosiglitazone/Metformin cooperative action in type 2 diabetes mellitus	46	−0.75	−2.73
Glycine, serine, cysteine and threonine metabolism	39	−0.77	−2.70

**Table 2 cells-11-00594-t002:** The most significantly regulated pathways from GSEA of ACEi compared with Veh-treated F1-Col4a3^−/−^ mouse kidneys.

Top 5 Pathways Downregulated by ACEi	Size	ES	NES
Wnt signaling in gastric cancer	59	−0.74	−2.33
Stem cells: embryonal epaxial myogenesis	45	−0.77	−2.32
Development: regulation of lung epithelial progenitor cell differentiation	61	−0.72	−2.32
Inhibition of oligodendrocyte precursor cell differentiation by wnt signaling	44	−0.77	−2.32
Deregulation of canonical wnt signaling in major depressive disorder	53	−0.74	−2.31
**Top 5 Pathways Upregulated by ACEi**	**Size**	**ES**	**NES**
Oxidative phosphorylation	77	0.84	2.84
Ubiquinone metabolism	38	0.84	2.56
Butanoate metabolism	24	0.90	2.48
Mitochondrial dysfunction in neurodegenerative diseases	91	0.70	2.42
Mechanism of Pioglitazone/Rosiglitazone/Metformin cooperative action in type 2 diabetes mellitus	46	0.78	2.40

**Table 3 cells-11-00594-t003:** The most significantly regulated pathways from GSEA of anti-miR-21 compared with Veh-treated F1-Col4a3^−/−^ mouse kidneys.

Top 5 Pathways Downregulated by Anti-miR-21	Size	ES	NES
Renal tubulointertitial injury in lupus nephritis	74	−0.71	−2.24
Development: regulation of epithelial-to-mesenchymal transition (EMT)	88	−0.67	−2.24
TH2 cytokine- and TNF-alpha-induced profibrotic response in asthmatic airway fibroblasts/myofibroblasts	63	−0.7	−2.21
Immune response: T-cell co-signaling receptors	62	−0.7	−2.21
WNT signaling in proliferative-type melanoma cells	60	−0.71	−2.21
**Top 5 Pathways Upregulated by Anti-miR-21**	**Size**	**ES**	**NES**
Oxidative phosphorylation	77	0.86	3.16
Mechanism of Pioglitazone/Rosiglitazone/Metformin cooperative action in type 2 diabetes mellitus	46	0.83	2.81
Ubiquinone metabolism	38	0.87	2.77
Mitochondrial dysfunction in neurodegenerative diseases	91	0.73	2.74
Butanoate metabolism	24	0.86	2.51

## Data Availability

Please send any additional requests to the corresponding author. RNA sequencing data are available from the Gene Expression Omnibus with submission number GSE179938 (https://www.ncbi.nlm.nih.gov/geo/query/acc.cgi?acc=GSE179938).

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
