# Peer review of "Anti-microRNA-21 Therapy on Top of ACE Inhibition Delays Renal Failure in Alport Syndrome Mouse Models"

_cells, 2022, doi:10.3390/cells11040594_

Round 1

Reviewer 1 Report

Rubel et al. report that anti-miRNA-21 therapy and ACE inhibition delay renal failure in the murine Alport syndrome model. This report shows the effectiveness of regulating microRNA expression on the current therapy (RAS inhibition) of Alport syndrome. Blocking miRNA-21 also affects the progression of genetic kidney disease.

I have minor comments on this works.

  1. Figures 1C and D show the levels of proteinuria and blood urea nitrogen. However, the script under X-axis is hard to read because of small. Authors need to change graphs (Figures1C and D) large enough to read.
  2. Figures 2 and 4 do not represent glomerular and tubular changes in the experimental groups. So, authors need to show higher magnification and show glomerular lesions and tubular lesions separately. 
  3. The authors need to check the legend of Figures 6 and 7. What does it mean “renal tubulointerstitial injury in lupus nephritis” and “mitochondrial dysfunction in neurodegenerative disease”? The authors need to describe more details in the methods sections.
  4. Figure 4 shows that anti-miRNA-21 treatment significantly decreases interstitial inflammation. So, authors need to present supporting data such as macrophage stain, inflammatory cytokine and chemokine expression by PCR or ELISA or Western blot, etc.
  5. What are the protective mechanisms in this study?

Author Response

Thank you for reviewing our manuscript. We believe that we can adress your points. Please see our written response.

Reviewer 2 Report

Thank you for the opportunity to read and review this article. The study seems well designed and well conducted. The results appear interesting and in line with the modern knowledge in the field.

Author Response

Thank you for reviewing our manuscript. Thank you for your positive feedback.

Reviewer 3 Report

Rubel and colleagues explored the therapeutic potential of anti–miR-21 oligonucleotides in combination or not with ACE inhibition in two Col4a3 deficient mouse models that mimic Alport nephropathy (a « fast-progressing » Col4a3-/- mouse model in the 129/SvJ background and an « intermediate progressing » mouse model in a mixed genetic background). In both models, miR-21 antagonism combined or not with ACE inhibition resulted in substantially milder kidney disease, with a better renal function and reduced histo-pathological features, compared with vehicle-treated mice. In particular, miR-21 inhibtition dramatically improved survival of Alport mice.

This manuscript is very well written and contains new data that could really be of potential interest to investigators in the challenging field of microRNAs and chronic kidney diseases. Moreover, these nice results reinforce interest for the ongoing HERA clinical trial that aims to evaluate the clinical potential of anti–miR-21 oligonucleotides.

However, the following issues need clarification.

1/ A schematic illustration of animal procedures should be added to make it easier to understand and compare both models

2/ How was urine collected ? using metabolic cages ?

3/ It would have been interesting to evaluate the renal expression of miR-21 in vehicle treated-mice in « fast-progressing » Col4a3-/- mouse model. Is miR-21 increase related to disease phenotype (fast or intermediate) ? Does ACEi treatment also reduce miR-21 renal expression in the « fast-progressing » Col4a3-/- mouse model ?

4/ ACEi monotherapy with ramipril was used at a lower dose in the « intermediate progressing » Alport model. This issue should be discussed more precisely.

5/ Both models seem to have been performed in two separate teams. It is particulat relevant since the overall protective effect of anti-miR 21 treatment is not affected by potential center bias. In the other hand, a large number of the investigated end-points are different, in particular concerning the histological features (laminin and fibronectin versus trichrome masson and alphaSMA). Then, both models comparison remains compromised. Would it be possible to implement the manuscript with a « common » marker ? If not, this point should also be discussed.

Author Response

(The authors gave the same response as above.)

Reviewer 4 Report

This paper aims to describe the use of ACE-inhibitor (ACEi) (in monotherapy) and in combination with anti-miR-21 therapy in two different mouse models of Alport syndrome. Those models corresponds to a fast-progressing and an intermediate-progressing model of the disease. Authors found significant improvements in several relevant markers of the disease in both, monotheraphy and when they combine both treatments. Moreover, the work also involves RNA sequencing of treated animals that demonstrates that the treatment is regulating the expression of several genes involved in pathways that are related to the progression of the disease.

The results described in the paper would be relevant for scientist working in Alport disease. Moreover these data support the use of this therapeutic approach in ongoing clinical trials in patients with this syndrome.

Overall, this manuscript is well written, the work has been well designed, the data are original, the methodology employed is adequate, and the results are properly analysed and described. In my opinion, this work is acceptable for Cells.

I have only some minor points to comment:

  • Page 2 line 15: Please define ACE for the first time in the text
  • Page 2 line 93: Please, define the purpose of using Ramipril, and the company name of the compound
  • Page 3 line 119, Page 6 line 258, Page 6 line 272, Page 8 line 362: use Anti-miR-21
  • Page 3 line 128: Please define BUN for the first time in the text
  • Page 5 line 213: The data are not available yet. Reviewers require a secure token
  • Page 5 line 242, Page 9 line 398 and 401: use “Ramipril”
  • Page 8 line 392 and 393: Use superscript for the -/- in Col4a3-/-
  • Page 14 lines 669-673 “The identities of these genes suggest that a better name for the pathway might be “renal tubulointerstitial fibrosis”, emphasizing the well-established role of TGFβ1 in stimulating the synthesis and excessive accumulation of extracellular matrix (ECM) proteins in mesangial cells, which in turn contribute to glomerulosclerosis”.  This paragraph does not reflect empiric results obtained in the studied, and perhaps should be included in the discussion section of the paper rather than in the results section.

Author Response

(The authors gave the same response as above.)
